# Differential Effects of a Telemonitoring Platform in the Development of Chemotherapy-Associated Toxicity: A Randomized Trial Protocol

**DOI:** 10.3390/diagnostics14060619

**Published:** 2024-03-14

**Authors:** Felipe Martínez, Carla Taramasco, Manuel Espinoza, Johanna Acevedo, Carolina Goic, Bruno Nervi

**Affiliations:** 1Centro Para la Prevención y Control del Cáncer (CECAN), Santiago 8331150, Chile; ctaramasco2@gmail.com (C.T.); manuel.espinoza@uc.cl (M.E.); johannaacevedo@udd.cl (J.A.); cgoicb@uc.cl (C.G.); bnervi@uc.cl (B.N.); 2Facultad de Medicina, Escuela de Medicina, Universidad Andrés Bello, Viña del Mar 2531015, Chile; 3Concentra Educación e Investigación Biomédica, Viña del Mar 2552906, Chile; 4Facultad de Ingeniería, Universidad Andrés Bello, Viña del Mar 2531015, Chile; 5Departamento de Salud Pública, Pontificia Universidad Católica de Chile, Santiago 8330023, Chile; 6Unidad de Evaluación de Tecnologías en Salud, Centro de Investigación Clínica, Pontificia Universidad Católica de Chile, Santiago 8330023, Chile; 7Instituto de Ciencias e Innovación en Medicina, Universidad del Desarrollo, Santiago 7550000, Chile; 8Facultad de Medicina, Escuela de Medicina, Pontificia Universidad Católica de Chile, Santiago 8330023, Chile; 9Foro Nacional del Cáncer, Santiago 8340696, Chile; 10Departamento de Hematología y Oncología, Escuela de Medicina, Pontificia Universidad Católica de Chile, Santiago 8330023, Chile

**Keywords:** telemonitoring, cancer, surveillance

## Abstract

**Simple Summary:**

This study aims to assess the impact of a telemonitoring platform on enhancing care for oncology patients undergoing chemotherapy. The research will conduct a randomized clinical trial involving recently diagnosed patients with solid carcinomas scheduled for curative intent chemotherapy. Participants will be divided into two groups: one using a smartphone application called *Contigo* for monitoring chemotherapy symptoms and providing cancer-related education, and another receiving standard in-person care. Patient experience during chemotherapy, severe chemotherapy-associated toxicity, quality of life, and user satisfaction with the application are among the measured outcomes. The study intends to enroll 80 participants and utilize various analytical methods, adhering to intention-to-treat principles.

**Abstract:**

Chemotherapy requires careful monitoring, but traditional follow-up approaches face significant challenges that were highlighted by the COVID-19 pandemic. Hence, exploration into telemonitoring as an alternative emerged. The objective is to assess the impact of a telemonitoring platform that provides clinical data to physicians overseeing solid tumor patients, aiming to enhance the care experience. The methodology outlines a parallel-group randomized clinical trial involving recently diagnosed patients with solid carcinomas preparing for curative intent chemotherapy. Eligible adult patients diagnosed with specific carcinoma types and proficient in Spanish, possessing smartphones, will be invited to participate. They will be randomized using concealed allocation sequences into two groups: one utilizing a specialized smartphone application called *Contigo* for monitoring chemotherapy toxicity symptoms and accessing educational content, while the other receives standard care. Primary outcome assessment involves patient experience during chemotherapy using a standardized questionnaire. Secondary outcomes include evaluating severe chemotherapy-associated toxicity, assessing quality of life, and determining user satisfaction with the application. The research will adhere to intention-to-treat principles. This study has been registered at ClinicalTrials.gov (NCT06077123).

## 1. Introduction

Cancer remains a significant global health challenge, being one of the leading causes of death worldwide. In 2020 alone, it accounted for nearly 10 million deaths, according to data from the World Health Organization [1]. This increase in cancer cases can be attributed to both the aging and growth of the population, accompanied by shifts in the prevalence and distribution of major cancer risk factors. Notably, many of these risk factors are interconnected with socioeconomic development. The global burden of cancer incidence and mortality is also on the rise at a rapid pace. The most prevalent forms of cancer include breast, lung, colon, rectum, and prostate cancers. In many cases, these forms of cancer will include chemotherapy as a treatment modality. According to a recent population-based study [2] drawing upon data from GLOBOCAN 2018, there is a notable surge in the demand for chemotherapy. Projections indicate a substantial 53% expansion in this demand by the year 2040. Although significant advances have been made in this field that directly improve overall survival of patients with these kinds of neoplasms, chemotherapy often results in a wide array of side effects such as nausea, pain, fatigue, and diarrhea, and in severe cases, they can even be life-threatening, as seen in instances of neutropenic fever and sepsis [3].

Systemic chemotherapy involves the delivery of drugs capable of inhibiting the replication of neoplastic cells [3]. It can be used with curative or palliative intent and in combination with other therapeutic measures for cancer treatment. There are multiple types of drugs with the ability to inhibit the cell cycle of neoplastic cells through various mechanisms, such as DNA alkylation (cisplatin, dacarbazine, busulfan), microtubule inhibition (docetaxel, paclitaxel, vincristine), or blocking nucleic acid synthesis by disrupting folic acid metabolism (methotrexate, cytarabine, gemcitabine), among others. Considering their mechanisms of action, the occurrence of adverse events is common both acutely and chronically, as these drugs also inhibit the cell cycle processes of healthy cells. The incidence of acute chemotherapy-associated toxicity has been estimated to range from 20% to 80% of patients undergoing this treatment, depending on the agents used; the clinical characteristics of the patients; and the method of detecting such toxicity, usually reported by the patients themselves [4,5,6,7]. Certain forms of toxicity are more common with specific agents, such as the cardiotoxicity of anthracyclines and the nephrotoxicity of platinum-derived chemotherapeutics (cisplatin, carboplatin) [3].

The adverse effects of chemotherapy can seriously impact the physical and mental health of patients, as well as their quality of life [7,8,9]. At times, these adverse reactions can lead to a reduction in exposure to chemotherapy agents, which in turn can also significantly impact the effectiveness of this form of treatment. Adverse events associated with chemotherapy are classified based on the National Cancer Institute method described in 2003, now referred to as the Common Terminology Criteria for Adverse Events. This method considers six grades of toxicity scored from 0 to 5, where grade 0 denotes the absence of such events, grade 1 is defined as mild, grade 2 is moderate, grade 3 is severe, grade 4 represents toxicity that is life-threatening, and finally, grade 5 is the one causing the patient’s death [10]. This classification can be categorized into two levels, where toxicity from grade 3 to 5 is high grade and from 0 to 2 is low grade [11,12]. On the other hand, there is growing evidence that patients receiving lower doses of chemotherapy may also see reduced chances of cure and survival compared to those not requiring such dose reduction [13,14,15,16].

In this regard, chemotherapy-associated toxicity is a highly relevant element for medical oncology teams. Its detection, characterization, and management are elements that can profoundly impact the treatment of cancer patients. Over the last few years, there has been significant and growing interest in developing alternatives that improve the detection of toxicity events for patients undergoing chemotherapy, as well as better tools that provide support for this patient population [17]. The effects of the COVID-19 pandemic have hindered access to in-person check-ups for patients, further amplifying the challenge for clinical healthcare teams to monitor the progression of patients and promptly detect potential adverse effects of chemotherapy [18]. Remote interventions, such as those based on phone check-ups [19,20] and mobile applications [21,22] have shown promising results in initial studies, but the current evidence weight does not allow for a definitive recommendation regarding their use. For this reason, this project has been designed to evaluate the feasibility of implementing a remote monitoring strategy and its effects on the clinical care of patients undergoing chemotherapy for solid neoplasms.

## 2. Objectives

The main objective of this study is to determine whether a telemonitoring platform that also provides clinical information to treating physicians responsible for patients with solid tumors improves the care experience for oncology patients. As secondary objectives, we aim to determine the ability to detect differences in the incidence of events attributable to chemotherapy-induced toxicity, as well as the frequency of hospitalizations, visits to emergency services, and reductions in chemotherapy doses.

Finally, a tertiary objective is to determine the level of satisfaction among the clinical team responsible for the patients regarding the information provided by the application.

## 3. Patients and Methods

To fulfill the aforementioned objectives, a randomized parallel-group clinical trial will be conducted among patients recently diagnosed with a solid carcinoma and preparing to initiate curative intent chemotherapy as part of their treatment at Hospital Sótero del Río and any of the centers belonging to the UC Christus Health Network. This protocol has been drafted following the guidelines of the Consolidated Standards of Reporting of Randomized Trials (CONSORT) [23] and the CONSORT-EHEALTH adaptation for web-based and mobile health interventions [24]. A flowchart of the study is presented in Figure 1.

(a)Participants

This study will include adult patients (aged >18 years, with no upper age limit) diagnosed with histologically confirmed lung, gastric, gallbladder, colon, breast, or cervical carcinoma, in any form or stage, who have initiated outpatient curative intent chemotherapy within the UC Christus Health Network or Hospital Sótero del Río facilities. Proficiency in the Spanish language and possession of a smartphone, regardless of the operating system (iOS^®^ or Android^®^), are also necessary for eligibility. Participant recruitment is scheduled to commence on 1 March 2024, and will continue until 30 December of the same year. All selected patients will be provided with an informed consent form to participate in the study. Exclusion criteria encompass individuals undergoing concomitant radiotherapy, those experiencing sensory impairment hindering the use of the application, individuals with cognitive impairment or psychiatric pathology preventing the use of the application, and those who express unwillingness to participate in the study.

Briefly, the recruitment process will proceed as follows. After confirming the cancer diagnosis by the treating team and defining the initiation of chemotherapy with the aforementioned characteristics, a message will be sent via email and/or a phone call to inform about the possibility of participating in the study (see call script in the Annex). This contact cannot be made within the first 72 h after the visit confirming the cancer diagnosis with the treating medical team. If there is an interest in participating in the study, a personal interview will be scheduled with study personnel (nursing technician or nurse) during the date of the next routine nursing check-up in the Oncology Unit of the participating hospitals. This visit corresponds to the education delivery control prior to the start of treatment. The healthcare team responsible for providing clinical services to patients will not be allowed to invite patients directly to the study. During the invitation, the interventions, potential benefits, procedures, and participation requirements will be explained, and a consent form will be signed.

(b)Procedures

Once informed consent is obtained, patients will be randomized in a 1:1 ratio using a permuted block method to receive one of the two intervention strategies. This procedure will be carried out by an investigator not involved in patient care using a computer algorithm that will be kept concealed from the other study investigators. Patients assigned to the active intervention group will receive a smartphone application called *Contigo* (see the *Application Design and Development* section, below). Briefly, this tool aims to fulfill two basic functions, which are monitoring the cancer patient for the early detection of signs and symptoms of oncology drug toxicity and delivering educational content that enables the patient to have tools to address common clinical situations associated with the diagnosis and treatment of their disease; for example, symptoms associated with chemotherapy, health plan coverages, and how to implement the treatment for a cancer patient. The first objective will be achieved through the deployment of modules and sub-modules, where the user will need to enter and record their perceptions and experiences through questionnaires associated with their oncological process. This monitoring will include a weekly search for chemotherapy toxicity-associated symptoms using questions derived from the Patient-Reported Outcomes of the Common Terminology Criteria for Adverse Events (PRO-CTCAE) questionnaire [25]. This tool was selected due to its public nature, making it widely used in oncology clinical trials, its standardization for detecting symptoms reported by patients, the possibility to select questions based on the toxicities to be detected, and its validation in the Spanish language [26]. In the case of detecting a severe toxicity event, the system will issue an alert to the treating team to contact the patient via phone and take the necessary measures for symptomatic control. Mild and moderate cases will receive educational information regarding general measures that can be implemented for their control. The information collected by the application will also be made available to the healthcare providers.

The second objective will be implemented through the delivery of educational health content for cancer patients. This educational content will be established by a team of professionals composed of medical oncologists, nurses, and oncology patients from the entities associated with the project through group sessions (Focus Group). In addition to this source for design, a documentary analysis of the updated scientific evidence and official reports or regulations from the Ministry of Health related to educational material for cancer patients will be carried out to complement the elements developed in the group sessions. The topics to be covered in this educational content will be specific to the type of cancer for each patient and will include aspects of the healthcare process, administrative aspects, health coverage, and self-awareness and self-care practices.

Those assigned to the control group will receive standard care and in-person check-ups as determined by their treating team. However, once the follow-up period for each participant in the study is completed (see below), all participants in the control group will be offered access to the application provided to the intervention group.

(c)Study Intervention: *Contigo* Application Design and Development

*Contigo* is an application that has been developed as part of a project funded with public resources (Agencia Nacional de Investigación y Desarrollo, ANID, FONDAP ID 152220002) led by PhD. Carla Taramasco, on behalf of the Universidad Andrés Bello, who holds the intellectual property rights to it. The concept behind *Contigo* emerged within a research proposal funded by the Pontificia Universidad Católica de Chile entitled “Identification and Measurement of Needs of Oncological Patients and Healthcare Professionals for the Development of New Technological Support Platforms for Patients”. The main objective of this project was to identify the information needs of both patients and healthcare professionals. To achieve this, focus groups and interviews were conducted.

Four breast cancer patients participated in the focus group, while nine healthcare professionals, including doctors and nurses who provide care to cancer patients and regularly participate in oncology committees, were involved in the interviews. The obtained results underwent a content analysis process to identify specific information needs. In response to these needs, the development of a mobile application for patients was proposed. The necessary software requirements were identified, and a technical design proposal for the application was developed, highlighting four main areas of information needs, including feedback for symptom recording during chemotherapy treatment, information about the disease, treatments, and tests from a process perspective, support information on administrative procedures of the national healthcare system, and additional information for the daily life of cancer patients.

Based on these elements, *Contigo* was developed as a Progressive Web App (PWA), utilizing web technologies that allow its installation on mobile devices, providing a user experience similar to that of a native application. This means that the application can be installed on any smartphone, regardless of the operating system it uses. The SCRUM agile software development methodology was applied to carry out the development of *Contigo*. At the end of each iteration, a project Quality Assurance team conducted functional tests to evaluate the quality of development and ensure user satisfaction. The application features a client–server architecture with modular design to facilitate scalability and simplify the incorporation of new functionalities. In its current version, *Contigo* includes six modules, including *El Viaje* (The Journey), *Mis Resultados* (My Results), *Mi Experiencia* (My Experience), *Asistencia* (Assistance), *Comunidad* (Community), and *Agendamiento* (Scheduling). A brief description of these modules is provided in Table 1, below, and the user interface depicted in Figure 2.

*Contigo* modules are continuously accessible to users throughout the day. Additionally, the application displays questionnaires according to the clinical trial schedule, meaning at 30, 60, and 90 days after randomization. These questionnaires are available through the application, and results can be viewed by the patient and clinicians using the same interface, as depicted in Figure 3.

The source code for *Contigo* will not be published. However, the project will have an informative website at www.contigoapp.cl, which will provide the community with images of the mobile application used in the study and flowcharts of the algorithms used, along with information on the results achieved with the implementation of the proposal. A functional demo version will also be made available to potential users to familiarize themselves with the application interface and its uses.

Patients participating in the clinical trial will receive the application free of charge. A username and password will be generated for each patient, and these will be provided by the monitoring nurse during the training on how to use the application. The monitoring nurse will also provide an introduction explaining the software’s usage. The research team in charge of the clinical trial will monitor the intervention, involving themselves only in patient training on the use of the *Contigo* application and providing technical support for any issues. The follow-up of oncological patients will continue as usual within the healthcare facility. In this way, the research team will only provide technical assistance, while healthcare professionals attending to the patient will provide clinical assistance.

(d)Variables

For each patient enrolled in the study, relevant personal information will be collected to interpret the study findings. The latter will be divided into demographic, household characterization, clinical, and neoplasia data. Demographic data will include information about gender, age, the highest level of education achieved, marital status (single, married, cohabiting), and the healthcare system (Fondo Nacional de Salud, FONASA, public sector; Institución de Salud Previsional, ISAPRE, private sector; or other). Household characterization data will consist of socioeconomic level, overcrowding, the availability of basic services, migratory status, and support networks. Lastly, the clinical information section will include data on height and weight, levels of creatinine and hemoglobin in the blood, as well as relevant medical comorbidities (diabetes mellitus, heart failure, chronic obstructive pulmonary disease, bronchial asthma, chronic liver disease, fibromyalgia, coronary heart disease, and chronic kidney disease), concurrent psychiatric conditions (depression, anxiety disorders, personality disorders), habits (alcohol consumption, substance use, or smoking), and neoplasia information, including the specific type of cancer, date of diagnosis, stage at diagnosis based on TNM classification (tumor, nodes, and metastasis), and the therapy received (monotherapy or combination therapy, standard or reduced dosage). Information regarding the patient’s functionality prior to the start of chemotherapy will also be included. For this purpose, the Eastern Cooperative Oncology Group Performance Status Scale (ECOG) will be used [31,32]. Briefly, the scale ranges from 0 to 5 points, where a score of 0 indicates a fully active patient capable of performing all daily activities without restrictions, while a score of 5 denotes a deceased patient.

Based on the aforementioned information, the risk of chemotherapy-related toxicity will be estimated. The Cancer Aging Research Group (CARG) score will be used to determine this risk. This score has been selected for its appropriate diagnostic capacity, its use of routinely collected data within a chemotherapy scheme, and its availability in multiple languages, including Spanish [33,34]. This score has been validated mostly in cohorts of elderly patients, and is also able to stratify toxicity risk by race [35]. The tool inquires about gender, height, weight, cancer type, chemotherapy dose to be received, agent scheme to be used, current functionality information, and hemoglobin and creatinine data. From these variables, a score ranging from 0 to 19 points will be estimated, where higher scores indicate a higher risk of developing systemic toxicity. The treating oncologist’s subjective impression regarding the risk of chemotherapy-related development will also be recorded. This impression will be quantified using a visual analog scale from 0 to 100 points, where the clinician will estimate their impression of the probability of developing a major toxic event attributable to chemotherapy [36].

(e)Outcomes

The primary outcome of this study is the experience lived by the patient during their chemotherapy treatment. To quantify this experience, the use of the OUT-PATSAT-35 questionnaire has been established, an adapted version of the European Organisation for Research and Treatment of Cancer OUT-PATSAT-35 questionnaire focused on evaluating the satisfaction of cancer patients regarding their healthcare [37,38]. This questionnaire contains 35 items referring to 12 multi-item scales divided into 3 sections [38]. The first two sections evaluate the medical and nursing staff in the chemotherapy-oriented version regarding their technical expertise, interpersonal skills, information delivery, and availability. The third section evaluates the department’s organization, the level of information exchange between healthcare providers (coherence, identification of the referring physician, etc.), interpersonal skills, as well as the level of information provided by the hospital team, waiting times, and the physical environment in which care is provided. The tool concludes with a general scale of overall satisfaction regarding the patient’s cancer treatment experience. All items receive a score based on a 5-point Likert scale, where the scores proportionally reflect the degree of satisfaction. Finally, all scores are linearly transformed into a scale of 0 to 100 points that aims to summarize the user’s experience into a single figure. This tool was selected for its psychometric abilities, as well as its availability, validation in the Spanish language, and the possibility of being independently applied by participants [30,39,40]. This outcome will be determined 3 months from the start of chemotherapy by the patient.

Among the secondary outcomes of this study is the development of severe chemotherapy-associated toxicity. To define severe events associated with chemotherapy, the Common Terminology Criteria for Adverse Events mentioned in the introduction were considered. Events of grades 3 to 5 were categorized as severe [11,12]. Other exploratory secondary outcomes consider the proportion of patients hospitalized due to chemotherapy-related toxicity, the number of visits to emergency services between groups, and the proportion of participants who received a reduction in their chemotherapy doses between groups. All secondary outcomes will be evaluated within the same time frames as the primary outcome.

The quality of life of the participants has also been considered as a second secondary outcome. For this purpose, the EQ-5D questionnaire will be used, measured 3 months from the start of chemotherapy by the participant. To carry out these measurements, the questionnaire will be administered electronically through a mobile application or via email. Participants will complete the questionnaire themselves, taking into account its psychometric properties, at the mentioned key follow-up moments [27,28]. In case electronic delivery of the questionnaire is not feasible, participants will be allowed to complete it during scheduled in-person follow-up visits.

Finally, the user satisfaction level of the clinical team using the application will also be considered among the secondary outcomes of this study. These outcomes will be exploratory and will allow for potential improvements in the system’s design for a subsequent clinical trial. Briefly, this exploration of clinician satisfaction relevant to the use of the application will be based on the responses to six statements [21] at the end of the study. Clinicians will express the frequency with which these statements interpreted their patient care experience based on a 5-point Likert scale, where scores of 1 point will mean “never”, 2 points “rarely”, 3 points “sometimes”, 4 points “often”, and 5 points “very often”. The statements are as follows:The information collected by the platform was useful for patient monitoring.The information collected by the platform improved communication with the patient.The information collected by the platform resulted in increased efficiency of patient care.The information collected by the platform improved the quality of patient care.I would like to use this platform to monitor future cancer patients.I would recommend this platform to my colleagues who monitor cancer patients.

Additionally, two open text spaces will be added where clinicians can express the perceived strengths and weaknesses of the platform for future developments. These fields will contain the following questions:–What aspects did you find good in your experience with the application?–What aspects need improvement within the platform?
(f)Statistical Analysis

### 3.1. Sample Size

Based on estimators from the literature [38,39], it was calculated that 80 participants (40 per group) will be required for this study. This estimation considers a between-group difference in the OUT-PATSAT-35 score of 10 points with symmetric standard deviations of 15 points between groups, a target statistical power of 80%, and standard significance levels (5% two-tailed alpha). The estimation also considers a possible 10% follow-up loss rate.

### 3.2. Analysis Strategy

The statistical analysis will initially employ means, standard deviation, and absolute and relative frequencies for descriptive analysis. For inferential analysis, Student’s *t*-test or Mann–Whitney U test will be used for comparing means based on data distribution and variances of the obtained data, and Fisher’s Exact Test for evaluating qualitative variables. Evaluation of primary and secondary outcomes that consider the time for event development (such as time to severe toxicity or hospitalization) will be carried out using Kaplan–Meier survival curves. Comparison between groups of these survival curves will be performed using the log-rank statistic. Additionally, potential associations will be quantified using the Hazard Ratio statistic in association with their respective 95% confidence intervals, allowing for better reporting in a subsequent larger-scale study. Subgroup analysis has not been considered in this study. All analyses will be carried out by a statistician unaware of the participants’ treatment assignment, following the intention-to-treat principle and using Stata v.16.0^®^ software (StataCorp LP, 1996–2020).

## 4. Conclusions

This study protocol describes a new alternative to address challenges in cancer care, particularly in the context of chemotherapy for various solid carcinomas. It introduces an innovative telemonitoring platform, the *Contigo* app, designed to monitor chemotherapy toxicity symptoms and provide educational content to cancer patients. This tech-driven solution is a response to the limitations exposed by the COVID-19 pandemic on conventional follow-up approaches, highlighting the need for remote monitoring in healthcare. *Contigo* aims at developing a patient-centric approach by prioritizing patient experiences during chemotherapy through validated questionnaires and embracing a multidisciplinary perspective. The application’s modules cover crucial aspects of cancer care, enabling patients to report their health status and experiences, access educational materials, and connect with community resources. Importantly, the study aims to not only enhance patient experience, but also evaluate the impact on healthcare providers, considering clinician satisfaction and its potential influence on patient care. While the protocol showcases robust data collection and statistical analysis plans, acknowledging certain limitations such as sample size and specific cancer types targeted, it provides a foundation for a promising study that could significantly advance cancer care by leveraging technology to bridge gaps in patient monitoring and support.

## Figures and Tables

**Figure 1 diagnostics-14-00619-f001:**
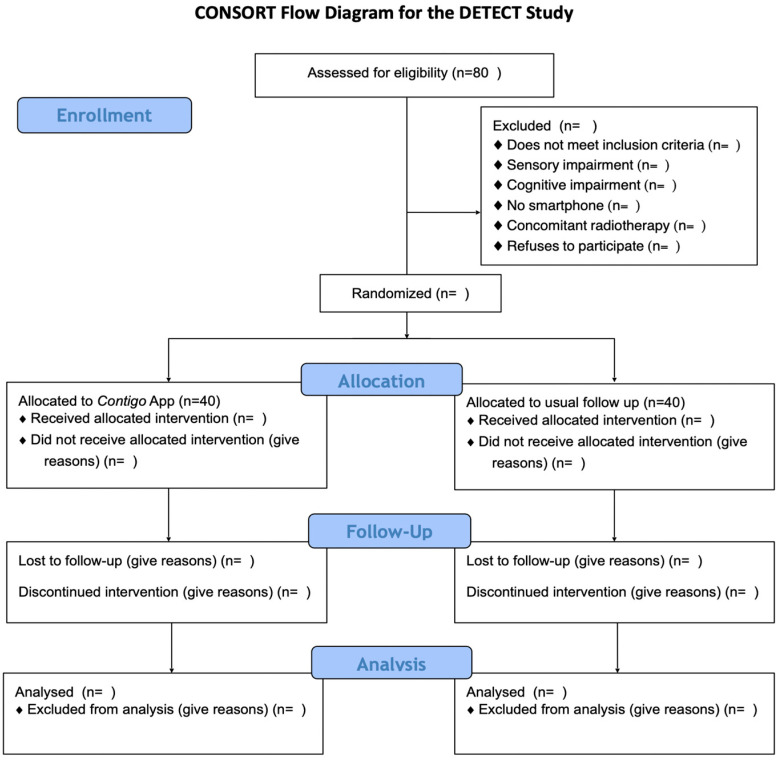
CONSORT flow diagram for the DETECT study.

**Figure 2 diagnostics-14-00619-f002:**
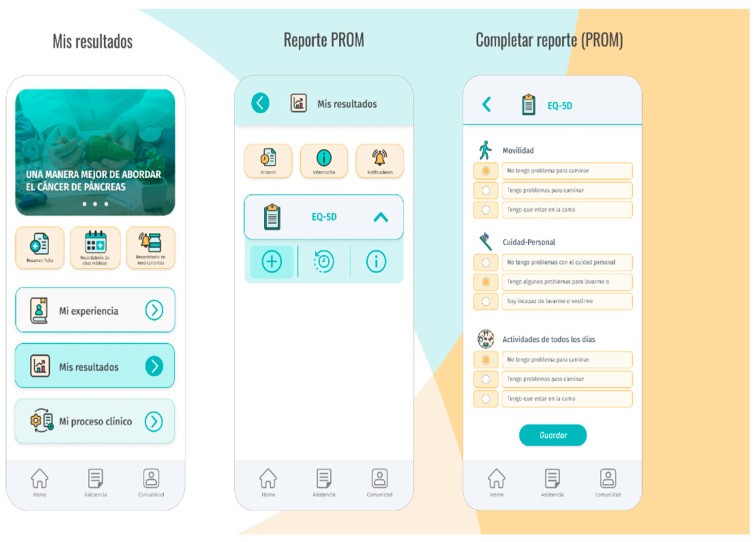
*Contigo* user interface.

**Figure 3 diagnostics-14-00619-f003:**
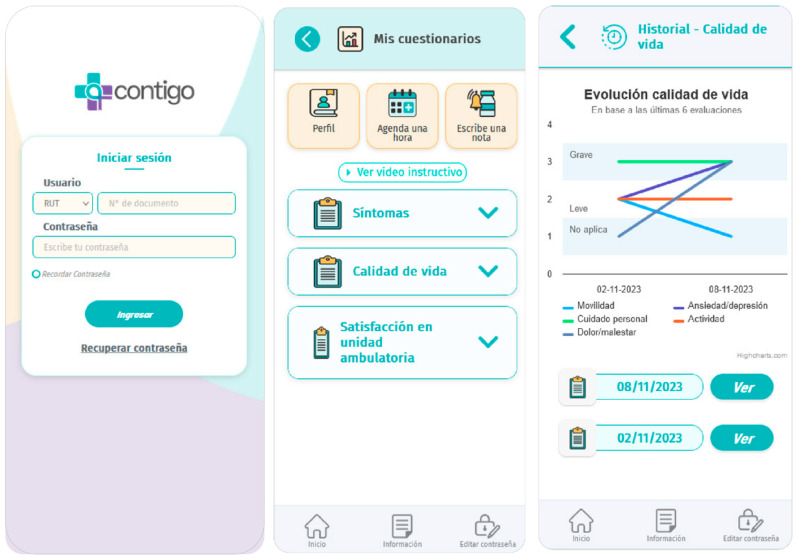
*Contigo* questionnaire modules.

**Table 1 diagnostics-14-00619-t001:** *Contigo* module description.

Module	Questionnaires	Description
The Journey	No	Provides information about the process of caring for cancer patients. It includes a straightforward description of the diagnostic and therapeutic procedures that will be implemented.
My Results	PRO-CTCAE [26] EQ-5D [27,28] PHQ-9 [29]	Allows for self-perception of health reporting through measures validated in Patient-Reported Outcome Measures (PROMs) and Patient-Reported Experience Measures (PREMs).
My Experience	OUTPATSAT-35 [30]	Allows for reporting experiences during healthcare and cancer treatment.
Assistance	No	Contains a list of frequently asked questions expressed by patients during cancer treatment.
Community	No	This module provides complementary information for non-clinical aspects that can help in daily life, such as where to obtain clothing, support groups, information and awareness activities, and others.
Scheduling	No	Information for requesting medical appointments and procedures.

Description of the *Contigo* application modules.

## Data Availability

No new data are currently available for data sharing. However, upon completion of the trial, data will be made available to researchers upon reasonable request by contacting the authors.

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
