# Peer review of "Differential Effects of a Telemonitoring Platform in the Development of Chemotherapy-Associated Toxicity: A Randomized Trial Protocol"

_diagnostics, 2024, doi:10.3390/diagnostics14060619_

Round 1
Reviewer 1 Report
Comments and Suggestions for Authors
The manuscript entitled “Differential Effects of a Telemonitoring Platform in the Development of Chemotherapy-Associated Toxicity: A Randomised Trial Protocol” The manuscript handles a good idea and is well-written. It explores telemonitoring as an alternative way emerged in sometimes that can help patients to connect with the health service team. However, I have some comments as follows:
Comments:
1. In line 125, the author mentioned that the study will be conducted from November 23 to July 24 on patients. How would the paper be published before ending the study?
2. Please highlight the paragraphs corresponding to the Chemotherapy-associated toxicity within the paper since it’s mentioned in the paper title.
3. I believe not all patients might have the technological capabilities to use this platform (ex. elderly people) to communicate with doctors. Please specify which age range you are targeting.
4. More information about the patients who participated in the study is required.
5. Minor punctuation mistakes.
Author Response
Dear Reviewer,
I extend my sincere gratitude for your feedback on our manuscript. Your constructive comments have enriched the quality of my work, and I appreciate the time and effort you dedicated to the review process.
Please find enclosed a response to the observations that were raised during the review process.
Kind regards,
Felipe Martinez on behalf of the authors.

Reviewer 2 Report
Comments and Suggestions for Authors
This randomized controlled trial protocol focuses on evaluating the effectiveness of telemonitoring in chemotherapy for solid tumor patients. The core of this research is a parallel-group randomized clinical trial involving patients newly diagnosed with solid carcinomas and preparing for curative intent chemotherapy.
The randomized controlled trial design is a strength, providing a robust framework for assessing the impact of telemonitoring. However, the reliance on smartphone technology and patient self-reporting through the app could introduce biases or inaccuracies in data collection, especially considering varying levels of technological literacy among patients, which has already been addressed by the research team.
The protocol is well-designed, well-presented, and suitable for publication in the current form.
Author Response
Dear Reviewer,
We appreciate your interest and kind words in considering our research protocol.
We agree with your perspective on the limitations of relying solely on self-reported symptoms and signs by participants to alert the clinical team to the potential development of chemotherapy-related toxicity. However, the use of these manifestations is still the customary way clinicians approach such diagnoses, and we have taken precautions to ensure that the quality of such reports is as reliable as possible.
We also thank your understanding of the inherent limitations of this situation, which will be duly included in the discussion of the work once completed.
Kind regards,
Felipe Martinez on behalf of the authors
Reviewer 3 Report
Comments and Suggestions for Authors
Dear authors,
I have reviewed your article " Differential Effects of a Telemonitoring Platform in the Development of Chemotherapy-Associated Toxicity: A Randomised Trial Protocol.
I think this manuscript is a very significant research protocol.
However, I think a few things need to be added, such as the CARG predictive tool is for older patients.
I have several comments.
1. “The Cancer Aging Research Group (CARG) score will be used to determine this risk. This score has been selected for its appropriate diagnostic capacity, use of routinely collected data within a chemotherapy scheme, and availability in multiple languages, including Spanish (33,34).”: You should cite this reference (PMID: 35565205) and mention that this tool also addresses the frequency of toxicity by race. In addition, you should mention that this tool was designed to predict solid tumor toxicity in patients over 65 years of age, and we do not know if it can predict solid tumor toxicity in patients younger than 65 years of age.
2. Reference 12: Oncologist. 2020 Oct;25(10):e1516-e1524.
Date of this review
19th Dec 2023
Author Response
Dear Reviewer,
Your time and valuable suggestions are greatly appreciated in our endeavor to enhance the quality of our manuscript. In the subsequent lines, we address each of your comments. Thank you once again for your constructive input; we are genuinely grateful for your expertise and guidance.
“The Cancer Aging Research Group (CARG) score will be used to determine this risk. This score has been selected for its appropriate diagnostic capacity, use of routinely collected data within a chemotherapy scheme, and availability in multiple languages, including Spanish (33,34).”: You should cite this reference (PMID: 35565205) and mention that this tool also addresses the frequency of toxicity by race. In addition, you should mention that this tool was designed to predict solid tumor toxicity in patients over 65 years of age, and we do not know if it can predict solid tumor toxicity in patients younger than 65 years of age.
We have added the suggested reference to the manuscript, specifying the risk by race. It was also noted that the CARG is primarily designed to assess chemotherapy-associated toxicity among older adults in the methodology section. While we anticipate that the majority of our patients will indeed be older adults, it is entirely possible that younger patients may be included in the study. The limitations of this approach will be acknowledged in the final manuscript in the study discussion. We have chosen to retain it as an option because it offers the best diagnostic accuracy and validation for use among patients in Chile.
Reference 12: Oncologist. 2020 Oct;25(10):e1516-e1524.
Thank you very much for the meticulousness in the review process of our manuscript. We have modified the reference as indicated. There was an error in the entry of the software we use for reference management, which has been corrected.
Kind regards,
Felipe Martinez on behalf of the authors
Round 2
Reviewer 1 Report
Comments and Suggestions for Authors
In line 132, the dates of the experiments are still not clear.
Author Response
Thank you for your comment.
We have improved the writing of the participants section to facilitate reading.
Kind regards,
Felipe Martinez on behalf of the authors.
Reviewer 3 Report
Comments and Suggestions for Authors
Please revise as follows;
Lines 281-282: This score has been validated mostly in cohorts of elderly patients (35), and is also able to stratify toxicity risk by race. ➝ This score has been validated mostly in cohorts of elderly patients, and is also able to stratify toxicity risk by race (35).
Author Response
Thank you for your observation. We have modified this sentence accordingly.
Kind regards,
Felipe Martínez on behalf of the authors.